# 3D Printing of Thermal Insulating Polyimide/Cellulose Nanocrystal Composite Aerogels with Low Dimensional Shrinkage

**DOI:** 10.3390/polym13213614

**Published:** 2021-10-20

**Authors:** Chiao Feng, Sheng-Sheng Yu

**Affiliations:** 1Department of Chemical Engineering, National Cheng Kung University, Tainan 70101, Taiwan; N36094136@gs.ncku.edu.tw; 2Core Facility Center, National Cheng Kung University, Tainan 70101, Taiwan; 3Program on Smart and Sustainable Manufacturing, Academy of Innovative Semiconductor and Sustainable Manufacturing, National Cheng Kung University, Tainan 70101, Taiwan

**Keywords:** polyimide, aerogels, 3D printing

## Abstract

Polyimide (PI)-based aerogels have been widely applied to aviation, automobiles, and thermal insulation because of their high porosity, low density, and excellent thermal insulating ability. However, the fabrication of PI aerogels is still restricted to the traditional molding process, and it is often challenging to prepare high-performance PI aerogels with complex 3D structures. Interestingly, renewable nanomaterials such as cellulose nanocrystals (CNCs) may provide a unique approach for 3D printing, mechanical reinforcement, and shape fidelity of the PI aerogels. Herein, we proposed a facile water-based 3D printable ink with sustainable nanofillers, cellulose nanocrystals (CNCs). Polyamic acid was first mixed with triethylamine to form an aqueous solution of polyamic acid ammonium salts (PAAS). CNCs were then dispersed in the aqueous PAAS solution to form a reversible physical network for direct ink writing (DIW). Further freeze-drying and thermal imidization produced porous PI/CNC composite aerogels with increased mechanical strength. The concentration of CNCs needed for DIW was reduced in the presence of PAAS, potentially because of the depletion effect of the polymer solution. Further analysis suggested that the physical network of CNCs lowered the shrinkage of aerogels during preparation and improved the shape-fidelity of the PI/CNC composite aerogels. In addition, the composite aerogels retained low thermal conductivity and may be used as heat management materials. Overall, our approach successfully utilized CNCs as rheological modifiers and reinforcement to 3D print strong PI/CNC composite aerogels for advanced thermal regulation.

## 1. Introduction

Polyimide (PI) aerogels have received wide attention because of their low density, stable mechanical properties, and low thermal conductivity [1]. Aerogels are highly porous materials for various applications, but they are often limited by their poor mechanical strength and thermal stability [2]. The capability of aerogels may be expanded by using engineering plastics such as PI with high mechanical strength and thermal stability [3]. For example, poly(4,4′-oxydiphenylene pyromellitimide), also known by its trade name Kapton, is a fully aromatic PI that retains its performance over a wide range of temperatures for application in aerospace engineering, defense, and electronics [4]. Recently, there has been a growing interest in using porous PI as high-performance aerogels [5]. Typical PI aerogels are made by the chemical imidization of crosslinked polyamic acids, followed by supercritical CO_2_ drying to remove organic solvents within the gels. The lightweight PI aerogels are especially attractive for thermal insulation [6,7], electromagnetic interference shielding [8], and aerospace engineering [9]. Nanocomposite PI aerogels can also be used as sensors [10] and nanogenerators [11]. However, the fabrication of PI aerogels is still limited to the traditional molding process. In addition, the large volumetric shrinkage of PI aerogels during preparation makes it difficult to control the shape of the final product precisely. It is desirable to achieve complex and customizable designs of PI aerogels to accurately assemble each component for advanced applications in thermal regulation and electronics.

Additive manufacturing (AM), also known as 3D printing, is an emerging tool for polymer processing. Through the layer-by-layer deposition of materials, geometrically complex objects from computer-aided design (CAD) can be achieved [12]. Light-based printing such as stereolithography (SLA) and digital light processing (DLP) utilized the photopolymerization of resins. On the other hand, fused deposition modeling (FDM) and direct ink writing (DIW) are extrusion-based printing. In particular, DIW is a low-cost and versatile technique compatible with a wide range of materials [13]. The DIW process prints the desired objects by extruding materials through fine nozzles. A successful layer-by-layer deposition is achieved by the proper viscoelastic response of the inks. The ink should display shear-thinning behavior for facile material extrusion under a high printing rate and high yield stress for good self-supporting ability [14,15].

Several strategies have been developed to 3D print PI-based materials. For example, photosensitive polyamic acids can be prepared by the reaction with common vinyl monomers [16,17,18]. Such inks can be used for SLA, DLP, and UV-assisted DIW. It is also possible to use the ionic interaction between polyamic acids and amine-containing vinyl monomers to prepare 3D printable inks for UV-assisted DIW and DLP [19,20]. The above approaches require the chemical modification of polyamic acids and a large quantity of organic solvents. The thermal imidization also induces a significant shrinkage of the printed object. Wang et al. demonstrated that DIW with a rapid drying apparatus enabled 3D printing of PI from poly(amic acid) ammonium salt (PAAS) hydrogels [21]. Water-soluble PAAS inks can be prepared by mixing polyamic acids and triethylamine (TEA). A careful selection of polyamic acids and the amines enables the formation of PAAS-based hydrogels through the potential hydrogen bonding and π–π stacking as physical networks [22]. In contrast to PI, the printing of PI aerogels has only been sporadically studied. It may be achieved by the freeze-drying of PAAS hydrogels [23] or using polymer microspheres as pore-forming agents and rheological modifiers for DIW [24].

Aside from the above methods, 3D printable ink may also be achieved by using nanoparticles as rheological modifiers. Recently, cellulose nanocrystals (CNCs) have been intensively studied because of their high elastic modulus and low density [25]. CNCs are rod-like particles (10 to 30 nm in diameter and few nanometers in length) with high crystallinity [26]. Indeed, CNCs have been included in chemically imidized PI aerogels [27] or PI thin films [28] as reinforcement. In addition, the dispersion of CNCs in water forms hydrogels with percolated networks. The CNC hydrogels exhibit strong shear-thinning behavior ideal for DIW [29,30]. Interestingly, it has also been demonstrated that the strong physical network of CNCs could significantly reduce the volumetric shrinkage of aerogels during the drying process [31].

Although several approaches have 3D printed PI-based materials successfully, the 3D printing of porous PI aerogels has only been sporadically studied. Furthermore, it is intriguing to utilize renewable nanomaterials such as CNCs to enhance the performance of PI aerogels further. In this work, we designed 3D printable inks to fabricate PI/CNC composite aerogels (Figure 1). The ink was mainly composed of water-soluble PAAS by the complexation of polyamic acids and TEA. CNCs were included in the aqueous solutions of PAAS to form physically crosslinked hydrogels for DIW. After printing, freeze-drying was employed as a simple and inexpensive method to make porous structures. Further thermal imidization of the freeze-dried PAAS/CNC aerogels leads to PI/CNC composite aerogels with improved mechanical strength and potential for thermal insulation. The physical network of CNCs effectively reduced the shrinkage of aerogels during preparation and enhanced the mechanical properties of the aerogels.

## 2. Materials and Methods

### 2.1. Materials

4,4-Oxydianiline (ODA, 98%) and triethylamine (TEA, 99%) were purchased from Acros Organics, Geel, Belgium. Pyromellitic dianhydride (PMDA, >98%) was obtained from Tokyo Chemical Industry, Tokyo, Japan. Cellulose nanocrystals (CNCs) were obtained from CelluForce Inc., Montreal, QC, Canada. The diameter and length of CNCs were 2.3 to 4.5 nm and 44 to 108 nm, respectively. N, N-Dimethylformamide (DMF, ACS grade) was offered by Avantor, Radnor, PA, USA, and was treated with molecular sieves for more than two days before synthesis. All other reagents were used as received.

### 2.2. Synthetic Procedures of PMDA-ODA Polyamic Acid

First, 4.806 g (24 mmol) ODA was added to a three-neck flask and mixed with 100 mL DMF under room temperature and a nitrogen atmosphere. A total of 5.235 g (24 mmol) PMDA was gradually added to form a polyamic acid precursor solution with a light yellow color. The solution was then stirred for another four hours to ensure the completeness of the reaction. Next, the viscous polyamic acid solution was diluted with another 100 mL DMF, and slowly poured into 600 mL deionized water under vigorous stirring for precipitation. The yellow polyamic acid was dried in a vacuum oven at 50 °C for more than two days before use. The detailed experimental procedure is summarized in Appendix A.

### 2.3. Preparation of PAAS/CNC Inks

First, 0.750 g polyamic acid, 0.363 g TEA, and 4.444 g deionized water were mixed in a glass vial. The solution was sonicated until all materials had dissolved to form the PAAS solution. Next, 700 mg of CNCs was dispersed in 4.444 g deionized water by a planetary mixer (MV-300S, CGT, Taiwan) at 1500 rpm for 10 min to form a uniform CNC hydrogel. Seven Teflon balls were added in the container to deagglomerate the CNC powder better. Finally, the aqueous PAAS solution was mixed with the CNC hydrogel by the planetary mixer at 1500 rpm for 30 min. The final concentration of polyamic acid and CNCs was 7.0 wt% and 6.5 wt%, respectively. For comparison, the amount of CNC was adjusted to make inks with 5.6 wt%, 6.5 wt%, and 7.4 wt% CNCs. Pure PI aerogels were prepared using a 7.5 wt% polyamic acid solution. Pure CNC aerogels were prepared from the inks containing only 6.5 wt% CNCs.

### 2.4. Rheological Test

The rheological behavior of inks was investigated by a rheometer (HR-2, TA instruments, New Castle, DE, USA) with parallel plates (diameter: 25 mm). The gap between the two parallel plates was set at 1000 μm. A steady-state shear flow test was used to measure the viscosity of the inks under different shear rates. The shear storage modulus (G′) and shear loss modulus (G″) were measured by an oscillatory stress sweep method at a frequency of 1 Hz. All rheological tests were performed at room temperature.

### 2.5. 3D Printing of PAAS/CNC Composite Ink

The as-prepared PAAS/CNC ink was transferred into a 10 mL disposable syringe. The bubbles within the ink were removed by the planetary mixer. A FDM printer (INFINITY X1Speed, INFINITY3DP, Kaohsiung, Taiwan) was modified for the DIW process by replacing the extruder with a 3D printed adapter. The syringe was then mounted on a syringe pump (NE-1010, New Era Instrument, Farmingdale, NY, USA) and connected to a 0.4 mm conical type plastic nozzle (TPND-22G-U, Musashi Engineering, Tokyo, Japan). The printer was controlled by Repetier software, and the G-codes were generated from the Slic3r toolpath generator. Typically, the printing speed was set at 20 mm/s, and the extrusion rate was 0.155 mL/min.

### 2.6. Preparation of PI, CNC, and PI/CNC Composite Aerogels

The printed objects were frozen in a −20 °C refrigerator overnight. The frozen samples were rapidly transferred to a freeze dryer and dried for more than two days. The dried samples further underwent a sequential temperature increase from 80, 120, 180, 240, to 300 °C under vacuum to form the PI/CNC composite aerogels. Each temperature was held for one hour, and the ramping rate was 1 °C/min in all cases.

### 2.7. Characterization

The compression strength was measured using a universal electromechanical testing machine (AGS-X 100kN, Shimadzu, Kyoto, Japan) equipped with a 10 kN load cell. All tests were performed following the ASTM-D695 standard.

The shrinkage ratio was calculated based on the following equation:$$Shrinkage\;\left(\%\right)=\;100\%\times \;\frac{V_{\mathrm{CAD}}-V_{s}}{V_{\mathrm{CAD}}}$$
where *V*_CAD_ represents the volume of the cylindrical CAD models and *V*_s_ is the apparent volume of the aerogels. Bulk density was calculated by the weight and volume of each sample after imidization.

Fourier transform infrared spectroscopy (FTIR) was performed using a Nicolet 6700 Fourier transform infrared spectrometer (Thermo Scientific, Waltham, MA, USA). The scanning region was set from 650 to 4000 cm^−1^. Resolution and scanning times were set at 2 cm^−1^ and 32 scans. Solid-state ^13^C NMR spectra were recorded using a Bruker Avance III HD 400 MHz NMR spectrometer. Thermogravimetric analysis (TGA) was conducted on a TGA 4000 (PerkinElmer, Waltham, MA, USA) from 30 °C to 700 °C under a nitrogen atmosphere at a heating rate of 10 °C/min. The microstructures of the aerogels were observed using a high-resolution field emission scanning electron microscope (SU-8010, Hitachi, Tokyo, Japan). The accelerating voltage was 10 kV.

Thermal conductivity (λ) was measured by Hotdisk TPS 2500S (Hot Disk AB, Gothenburg, Sweden) with a 5465 f1 type Kapton^®^ sensor and standard isotropic method under room temperature. The probing depth, heating power, and measuring time were set at 5 mm, 10 mW, and 10 s, respectively. Thermal images were taken by a FLIR C2 thermal imaging camera (FLIR, Wilsonville, OR, USA). The working distance was around 30 cm. Samples were placed on a 160 °C hotplate. A series of thermal images were taken before and after 10 min of heating treatment.

## 3. Results and Discussions

### 3.1. Rheological Analysis of the PAAS/CNC Composite Inks

We first explored the effect of CNCs on the rheological behavior of the PAAS/CNC inks. The concentration of CNCs varied from 5.6 wt% to 7.4 wt%, and polyamic acid was 7.0 wt%. Water-soluble PAAS was prepared by mixing PAA with TEA. TEA also catalyzes the imidization reaction of PAA to PI [32]. As mentioned earlier, CNCs form a reversible physical network in water, and a series of rheological analyses can evaluate the strength of the network. All inks demonstrated significant shear thinning behavior (Figure 2a). For example, the viscosity for the ink using 6.5 wt% CNCs was 18,134 Pa∙s at 0.01 1/s, but it decreased to 228 Pa∙s at 1 1/s. The reduction in viscosity at a high shear rate allows for the ink to be extruded at a high flow rate. The performance of DIW also depends on the storage modulus and yield stress. The ink should have a high storage modulus and yield stress, so the printed inks could stack successfully and even span through unsupported parts [33]. The yield stress of the ink could be obtained from the intersection between the storage (G′) and loss modulus (G″). As shown in Figure 2b, all inks exhibited gel-like behavior with their storage modulus greater than loss modulus at small shear stress. The gel-like behavior of the inks at low shear stress was found in several works by studying the suspension of CNCs [29,30,34]. CNCs form a percolating network in ink with finite yield stress. The ink does not flow if the applied stress is lower than the yield stress. At high shear stress, the inks transited into a liquid-like state (G″ > G′) suitable for DIW because of the alignment of CNCs [34]. In addition, both storage modulus and yield stress increased with the concentrations of CNCs. Therefore, the strength of physical networks in the inks gradually increased with CNCs.

From the above result, the inks using 6.5 or 7.4 wt% CNCs were sufficient for DIW because of their high yield stress (>200 Pa) [15]. Our previous work has shown that a higher concentration of CNCs (12 to 15 wt%) in water was needed to achieve similar rheological performance for DIW [35]. Therefore, PAAS may promote the formation of the physical network by CNCs and reduce the concentration of CNCs needed for printing. Similar behavior has been found for the dispersion of CNCs with anionic polymers such as carboxymethyl cellulose [36]. The rheological behavior of CNCs is strongly influenced by polymers, especially non-adsorbing polymers that have little affinity with the surface of CNC particles [37,38]. The presence of non-adsorbing polymers significantly reduced the gelation concentration of CNCs. This phenomenon can be attributed to the depletion force induced by the non-adsorbing polymers. The polymers lose their conformation entropy around the CNC particles and formed polymer-depleted regions near CNCs. Therefore, the difference in osmotic pressure between the depleted region and the bulk polymer solution provides entropically driven attraction to promote the gelation of CNCs. We hypothesized that the depletion-induced flocculation of CNCs also occurred in our PAAS/CNC inks. PAAS is an anionic polymer [22,32] that should have little adsorption on negatively charged CNCs. Therefore, PAAS may have a similar behavior to the non-adsorbing polymers [36]. The depletion effect of PAAS then reduced the concentration of CNCs required for DIW.

The effect of polyamic acid on the rheological behavior of the inks has also been investigated. A CNC of 6.5 wt% and the concentrations of polyamic acid ranged from 4.7 wt% to 9.7 wt%. All three samples had similar shear-thinning behavior for DIW (Appendix A). Appendix A further shows these inks had a similar response in storage modulus and yield stress. Although the presence of non-adsorbing polymers promotes the gelation of CNCs, the effect was less prominent at a high concentration of polymers [36,39]. Based on the above analysis, we concluded that adding more polyamic acids into the system did not significantly improve the printability of the ink.

The above results suggest that the viscosity and modulus were mainly determined by the concentration of CNCs rather than polyamic acids. A 6.5 wt% of CNCs was sufficient to achieve significant shear thinning behavior and high yield stress for DIW. Based on the above analysis, we then mainly focused on the effect of CNCs on the properties of the PI/CNC composite aerogels. We chose the ink with 6.5 wt% CNCs to demonstrate the printing performance (Figure 3). During the printing process, the ink could be successfully extruded and stacked layer-by-layer (Figure 3a,b). The printed object then underwent the freeze-drying process to remove water and form macropores. Further thermal treatment completed the PI/CNC composite aerogels. As shown in Figure 3d, the thermal imidization induced only small dimensional shrinkage. The printed object showed good shape-fidelity compared to our initial CAD design. Some cracks appeared in the imidized samples, possibly because of the drastic weight loss of the composite aerogels or thermal stress [40] during the thermal imidization. The presence of cracks may be mitigated by further adjusting the imidization process. From scanning electron microscopy (SEM), we found that the printed PI/CNC aerogels had high spatial resolution as expected (Figure 3e), and small pores could be observed on the surface of the aerogels (Figure 3f). Furthermore, a cylindrical model with a high aspect ratio was printed using the PAAS/CNC ink (Appendix A). The ink could successfully be stacked up to 70 layers (28.22 mm in height), indicating the high yield stress and the printability of the PAAS/CNC ink.

### 3.2. Chemical Analysis of the PI/CNC Nanocomposite Aerogels

The PI/CNC aerogels were analyzed by FTIR to probe the degree of imidization (Figure 4a). The PI aerogel without CNCs and the pure CNCs aerogel were used for a comparison. Pure PI aerogels had two peaks at 1776 cm^−1^ and 1721 cm^−1^, corresponding to the asymmetrical and symmetrical C=O stretching of the imide ring, respectively [8]. The peaks at 1500 cm^−1^ and 1375 cm^−1^ came from the C=C stretching of the aromatic ring and the C–N stretching of the imide ring. Besides these characteristic peaks of PI, the absence of signals at 1540 and 1660 cm^−1^ (amide II and C=O stretching of amide) also implied a successful imidization of the pure PI aerogels. As shown in Appendix A, the pure CNC aerogel had a broad peak around 3400 cm^−1^, corresponding to the hydroxyl groups on CNCs. The FTIR spectrum of the PI/CNC aerogels was similar to that of the pure PI aerogel, suggesting that the imidization was not affected by CNCs. Mixing CNCs with PAAS did not affect the final cyclization process of PAAS to PI. In addition, the PI/CNC aerogels also showed the characteristic peak of the hydroxyl group at 3400 cm^−1^. Therefore, a significant amount of CNCs remained in the composite aerogels, even after the thermal imidization process.

Solid-state ^13^C NMR was also used to probe the interaction or chemical bonding between PI and CNCs in the composite aerogels (Figure 4b). The characteristic signals of CNCs were found at 65.6 ppm (C6 crystalline), 70–75 ppm (C2, C3, C5), 84–89 ppm (C4), and 104.9 ppm (C1) [41]. In both pure PI and PI/CNC aerogels, we found the peaks of C=O imide moiety (165.6 ppm), phenoxy carbon (156.2 ppm), and the peaks of aromatic rings (115–134 ppm). These signals were similar to early reported results [42,43]. The peaks around 170–190 and 70–90 ppm were attributed to the spinning sidebands by the magic angle spinning [44]. The main signals of CNCs could also be found in the composite aerogels, suggesting that most CNCs were not degraded by thermal imidization. In addition, esterification might also occur between the carboxylic acid groups of PAA and the surface hydroxyl groups of CNCs during the thermal treatment. As we did not find any new signals in the PI/CNC composite aerogels, the reaction between PAA and CNCs should be minimal and the CNCs mainly acted as rheological modifiers and reinforcing agents. The above results also suggest that complete imidization was achieved in the composite aerogels and a significant amount of CNCs still remained.

Besides FTIR and solid-state NMR, we also used TGA to confirm the completeness of the thermal imidization process (Figure 4c). Aromatic polyimides typically possess great stability in a high-temperature environment. For the pure PI aerogel, no significant weight loss was found until 550 °C, and the final char yield was 51%. The onset temperature of degradation was approximately 576 °C, indicating a high-temperature tolerance of the pure PI aerogel. On the other hand, the pure CNCs aerogel decomposed around 290 °C and showed a char yield of 7 wt% at 760 °C, similar to the previous report [27]. The PI/CNC composite aerogels had two degradation temperatures around 300 °C and 540 °C. The first onset of decomposition at 300 °C was the degradation of CNCs, and the second one at 540 °C should be the thermal decomposition of PI. We also found that the thermal stability of CNCs increased slightly in the PI/CNC aerogels, but the decomposition temperature of PI decreased slightly. The improved thermal stability of CNCs in the composite aerogels may come from the hydrogen bonding between the surface hydroxyl group of CNCs and PI [45]. Although CNCs may degrade around the thermal imidization temperature (300 °C), a large quantity of CNCs still remained in our PI/CNC aerogels based on the results of TGA. We then recorded the weight of each freeze-dried sample before and after the thermal imidization process to understand the degradation behavior of CNCs (Appendix A). For the pure PI aerogel without CNCs, a significant loss of weight to 42.3% was observed. This result could be attributed to the loss of water by imidization and the evaporation of TEA. If a similar degree of weight loss occurs in the composite aerogels, the 6.5 wt% CNCs sample should only lose 25.8% of its weight. However, the observed weight loss was 32.4%, indicating that some CNCs degraded during the thermal imidization process. The degradation of CNCs was more significant in the aerogels with 5.6 wt% CNCs than the other composite aerogels. Fortunately, a significant amount of CNCs remained in the aerogels prepared by 6.5 wt% and 7.4 wt% CNCs. Based on the above results, we expect that the weight loss of CNCs should be minimal, even at 300 °C if there is a high concentration of CNCs. This result may come from the enhanced thermal stability of CNCs in the composites, as indicated by our TGA analysis. Although lowering the imidization temperature could preserve CNCs in the composite aerogels, the mechanical properties of PI may be compromised [46].

### 3.3. Internal Morphology of the PI/CNC Nanocomposite Aerogels

We further analyzed the internal morphologies of the aerogels prepared using different concentrations of CNCs. Figure 5a,c shows that the PI/CNC aerogels prepared by 6.5 wt% CNCs had a lamellar structure. This morphology was consistent with the previous work studying CNC aerogels prepared under a slow freezing rate [47]. The paralleled pores came from the anisotropic growth of ice crystals along the freezing front. The pure PI aerogels generally showed a smaller pore distribution, but the lamellar morphology was less pronounced. In addition, the morphology of the aerogels using different concentrations of CNCs was similar (Appendix A), indicating that the presence of CNCs was sufficient to form the lamellar structure. We also examined the morphology of the freeze-dried pure PAAS and pure CNC aerogels (Appendix A). The PAAS aerogels also contained some parallel sheet-like structures, indicating that anisotropic growth of ice crystals still occurred. However, such structures generally disappeared after thermal imidization, potentially because of the shrinkage of the aerogels. On the other hand, the freeze-dried CNC aerogels formed pores much larger than the composite aerogels. This result was expected as a low concentration of CNCs was not enough to form strongly packed networks. The pure CNCs aerogel was also too fragile for further studies. The results of the microstructures further support our rheological analysis. The physical network of CNCs was facilitated by the presence of PAAS in the inks. Although the pure PI aerogels contained smaller and denser pores than the composite aerogels, the lamellar structures induced by CNCs may further enhance the thermal insulation performance of the aerogels. Some studies have attempted to control the growth of ice crystals to modulate the thermal insulation of aerogels in different directions [6,9]. We are currently investigating the possibility of integrating 3D printing and directional freezing.

As implied in the internal morphologies, CNCs should significantly reduce the shrinkage of the aerogels during preparation. The degree of shrinkage is summarized in Figure 6a. As the concentration of CNCs increased, the volumetric shrinkage generally decreased, suggesting that the aerogels had higher shape-fidelity after imidization. The PI/CNC composite aerogels only had about a 10% decrease in volume after freeze-drying. Further imidization reduced the volume of the composite aerogels, but they still retained more than 70% of the initial volume.

In contrast to the composite aerogels, the pure PI aerogels had a much higher shrinkage ratio, up to 60%. These results demonstrate that CNCs could strengthen the polymer matrices during freeze-drying and thermal imidization. As shown in Figure 6b, the pure PI aerogel shrank and wrapped significantly after thermal imidization. In contrast, the composite aerogel and the pure CNC aerogel only shrank slightly. However, the color of the pure CNC aerogels became darker, indicating that some CNCs were thermally degraded. Appendix A shows the result of the bulk density. The bulk density of the composite aerogels increased with the concentration in CNCs because of the increased solid content. The severe shrinkage of the pure PI aerogel also led to a high bulk density.

### 3.4. Performance of the PI/CNC Composite Aerogels

The mechanical properties of the PI/CNC composite aerogels were studied, as shown in Figure 7. All samples followed the typical compression stress–strain curves of aerogels including an elastic zone, a platform zone, and a dense zone [48]. The pure PI aerogel had the lowest compression modulus. On the other hand, the compression modulus and maximum stress increased with the concentrations of CNCs. In general, the modulus of aerogels increases with their density [49]. As mentioned earlier, the pure PI aerogels generally had smaller and denser pores than the composite aerogels because of the shrinkage during thermal imidization. As a result, the density of the PI aerogel was similar to the composite aerogels. Under a similar bulk density, CNCs greatly reinforced the aerogels during compression. Although the pure PI aerogels contained dense and small pores that are generally desirable for strong aerogels, the effect of CNCs was more significant. As shown in Figure 7b, our PI/CNCs demonstrated a high compression modulus ranging from 6.40 to 13.64 MPa. The mechanical stiffness was higher than the previous works using a lower concentration of nanofillers [6,8,48]. Therefore, a sufficient concentration of CNCs provides both 3D printability and high mechanical strength.

Finally, we compared the thermal insulation ability of the aerogels. Figure 8a shows a series of thermal images for pure PI, pure CNCs, and three composite aerogels with a cuboid shape. These aerogels were placed on a hot plate at 160 °C. After 10 min, we found that all samples retained a temperature below 80 °C, suggesting they all had low thermal conductivity. The concentration of CNCs appeared to have minimal effects on the thermal conductivity of the aerogels based on the thermal images. Indeed, the thermal conductivities of the composite aerogels with 5.6 wt% and 6.5 wt% CNCs were 89.39 ± 0.087 and 92.27 ± 1.03 mW/m∙K, respectively. This result was comparable to the thermal conductivities of the PI aerogels prepared by the supercritical CO_2_ drying [50] and bidirectional freezing [6]. We also attempted to measure the thermal conductivity of the pure PI, pure CNCs, and the composite aerogels with 7.4 wt% CNCs. However, the accurate measurement could not be achieved because these samples typically had low mechanical integrity or high surface roughness.

In addition to common cuboid structures, we also placed a 3D printed pyramid on the hotplate (Figure 8b). The pyramid was 1 cm in height with 30 printed layers, indicating the robust printability of inks. We found that there was a gradient distribution of temperature from the hot plate to the top of the printed pyramid. Therefore, the 3D printed sample still had good thermal insulation ability, as expected. A grid with a finer surface structure than the mesh in Figure 3 was also printed for the thermal image analysis. As shown in Figure 8b, the temperature distribution of the grid depended on the macrostructure enabled by 3D printing. Therefore, 3D printing could provide an opportunity to locally control the temperature distribution and modulate the overall thermal conductivity of the aerogels. Although some effective designs could also be fabricated by molding, 3D printing is a rapid test of different macrostructures. The combination of thermal insulation capability and 3D printability of our PI/CNC composite aerogels may enable the easy production of a customizable design for advanced applications.

Our results indicate that CNCs can be used as renewable fillers to 3D print PI/CNC composite aerogels with enhanced mechanical strength. Although our PAAS/CNC ink was printed under room temperature, several previous works preheated the inks to reduce the viscosity of the ink, facilitate material extrusion, and extend the printable time [21,23,51]. The printing speed and resolution may be further improved by selecting a suitable heating parameter. Further work is being conducted to understand the effects of temperature and heating time on the rheological behavior of the PAAS/CNCs inks. In addition, DIW can be used to align CNCs by controlling the shear stress during printing [29,34]. The aligned CNCs in the composites exhibited anisotropic mechanical response and optical properties. The alignment of nanomaterials can also be achieved by high-resolution aerosol jet printing [52]. Therefore, the mechanical stiffness and thermal insulating properties of the PI/CNC aerogels may also be controlled by the printing process.

The low degradation temperature of CNCs limits the overall thermal stability of the aerogels compared to PI. The thermal resistance of the PI/CNC aerogels may be improved by the surface modification of CNCs [53,54]. In addition, the thermal insulation capability of the PI/CNC aerogels was affected by the distribution and the direction of pores [6]. Although our work only showed the 3D printability of the PI/CNC composite aerogels, we envision that a further combination of 3D printing with other pore-forming strategies may produce robust PI-based aerogels with enhanced thermal regulation performance.

## 4. Conclusions

In this study, we utilized CNCs as renewable nanofillers to prepare water-based inks for the 3D printing of PI composite aerogels. Freeze-drying and thermal imidization of the printed objects produced porous PI/CNC composite aerogels. The potential depletion effect by PAAS promoted the gelation of CNCs and reduced the required concentration of CNCs for DIW. In addition, the strong physical networks of the CNCs increased the thermal stability of the porous structures and reduced the shrinkages of the aerogels after thermal imidization. Furthermore, CNCs reinforced the PI/CNC composite aerogels to provide high mechanical strength. Our composite aerogels exhibited low thermal conductivities for thermal regulation. Overall, our work demonstrated a simple way to print PI aerogels with customizable shapes, low density, and high mechanical strength. The 3D printing of high-performance composite aerogels may further enable advanced and rapid design of thermal insulating materials to precisely control the local temperature distribution of the target device.

High-resolution printing by DIW can be achieved by using the ink containing only 6.5 wt% CNCs.Chemical analysis showed that a significant amount of CNCs remained in the composite after thermal imidization, and CNCs displayed improved thermal stability.CNCs significantly reduce the dimensional shrinkage of the aerogels during preparation and enable the 3D printing of aerogels with high shape fidelity.The PI/CNC composite aerogels demonstrated increased mechanical stiffness and sufficient thermal insulating capability.

## Figures and Tables

**Figure 1 polymers-13-03614-f001:**
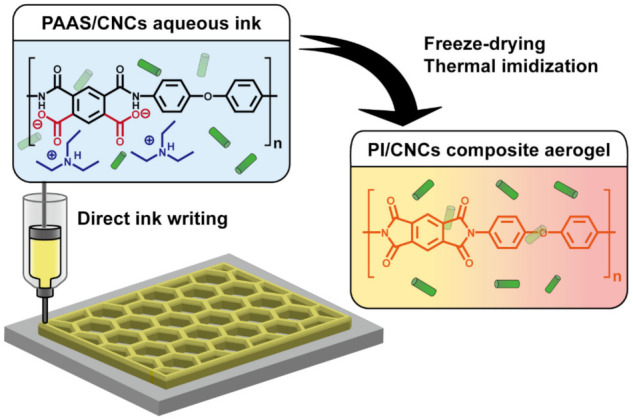
Synthesis and 3D printing of PI/CNC composite aerogels.

**Figure 2 polymers-13-03614-f002:**
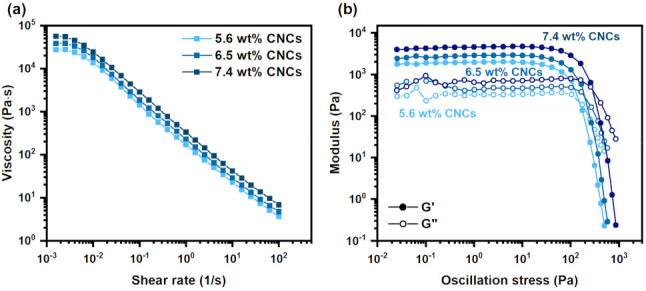
Rheological analysis of the PAAS/CNC inks under different concentrations of CNCs. (**a**) Steady-state shear viscosity of the inks. (**b**) Oscillatory rheological measurement.

**Figure 3 polymers-13-03614-f003:**
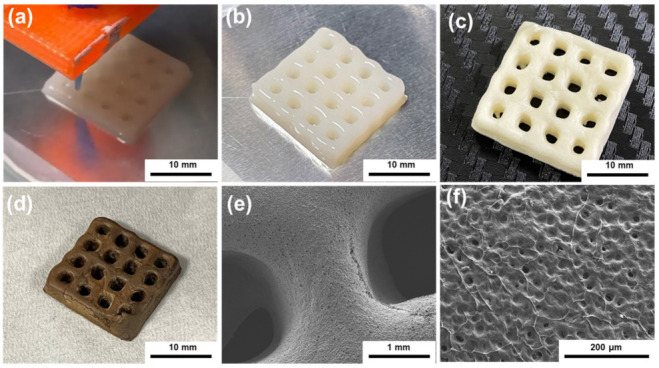
Performance of the PAAS/CNC ink for DIW. (**a**) The printing process. (**b**) The printed mesh structure. (**c**) The printed mesh after freeze-drying (**d**). The printed mesh after thermal imidization. (**e**,**f**) SEM images of the mesh after thermal imidization under different magnification.

**Figure 4 polymers-13-03614-f004:**
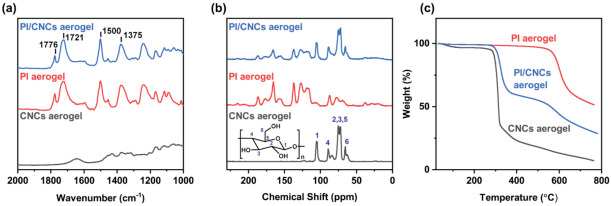
(**a**) FTIR spectra of the aerogels. (**b**) Solid-state ^13^C NMR spectra of the aerogels. (**c**) TGA curves of aerogels. The PI/CNC aerogel was prepared by using 6.5 wt% CNCs.

**Figure 5 polymers-13-03614-f005:**
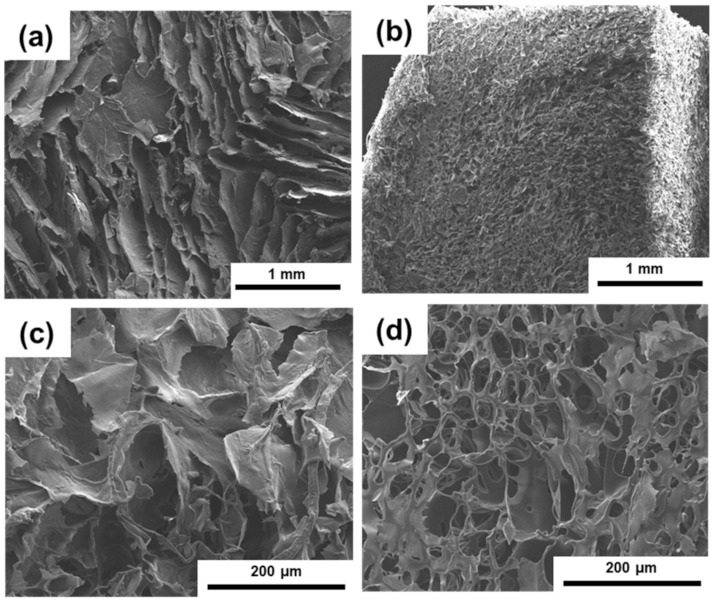
Cross-sectional SEM images of the aerogels after the thermal imidization process. (**a**,**c**) The PI/CNC composite aerogel using 6.5 wt% CNCs. (**b**,**d**) Pure PI aerogel.

**Figure 6 polymers-13-03614-f006:**
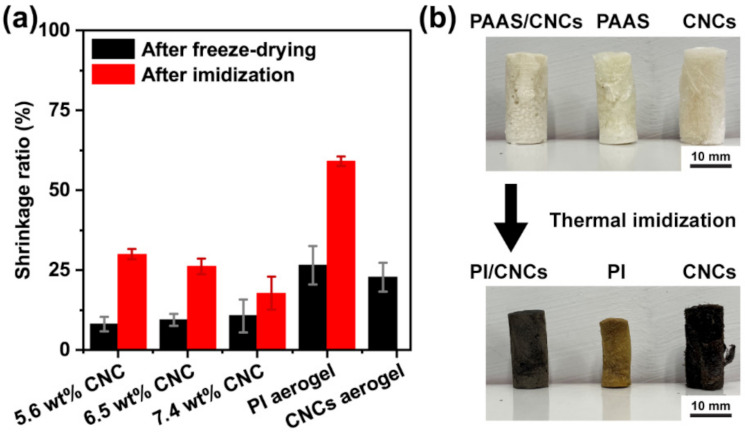
(**a**) Volumetric shrinkage ratio of aerogels after freeze-drying and after thermal imidization. (**b**) The appearance of aerogels before and after thermal imidization.

**Figure 7 polymers-13-03614-f007:**
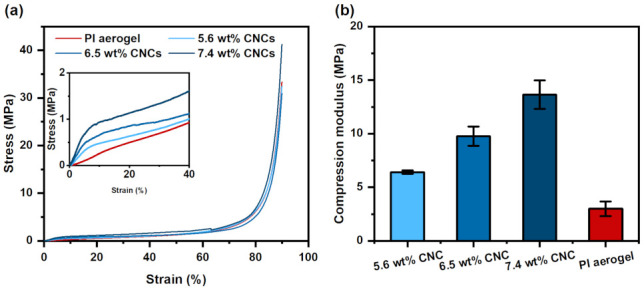
Mechanical properties of the aerogels using different concentrations of CNCs. (**a**) Compressive stress–strain curves. (**b**) Comparison of compressive modulus.

**Figure 8 polymers-13-03614-f008:**
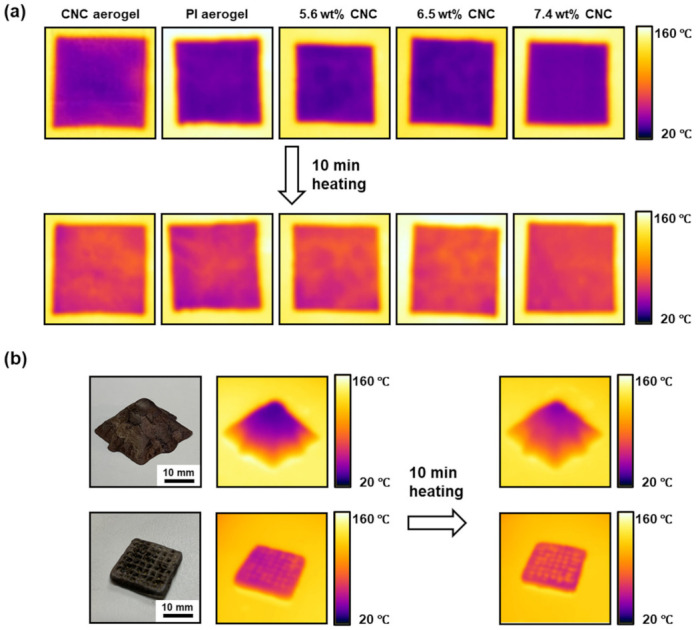
The thermal insulation performance of aerogels. (**a**) Infrared images of pure CNC aerogel, pure PI aerogel, and PI/CNC aerogels with different concentrations of CNCs. The sample size was 2.8 × 2.8 × 0.4 cm^3^. (**b**) 3D printed pyramid and grid of PI/CNC aerogels.

## Data Availability

The data presented in this study are available on request from the corresponding author.

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
