# Peer review of "3D Printing of Thermal Insulating Polyimide/Cellulose Nanocrystal Composite Aerogels with Low Dimensional Shrinkage"

_polymers, 2021, doi:10.3390/polym13213614_

Round 1
Reviewer 1 Report
The authors successfully 3D printed PI/CNC composite aerogel, which is interesting. However, some more data are needed to beef up claims.
There is a lack of explanation and demonstration why 3D printing is needed. The designs shown in Figure 3, Figure 6, and Figure 8 are easy to cast without using DIW. It is necessary to demonstrate a useful aerogel part that can be fabricated only by 3D printing.
The authors did not study the thickness. What is the maximum thickness allowable for the authors’ method?
Figure 3d. After thermal immunization, there appear to be some cracks. Why?
The observation of increased thermal stability has also been reported in other polymers. The study below is relevant and could be useful to extend the discussion.
- Cui Y, Jin R, Zhang Y, et al., 2021, Cellulose Nanocrystal-Enhanced Thermal-Sensitive Hydrogels of Block Copolymers for 3D Bioprinting. Int J Bioprint.
Sometimes preheating can improve DIW printability (see below). Could the authors comment if this could help their material? Please include this in discussion.
- Tan JJY, Lee CP, Hashimoto M, Preheating of Gelatin Improves its Printability with Transglutaminase in Direct Ink Writing 3D Printing, Int J Bioprint, 6(4): 296.
Though aerogel is not aerosol, like many, I am curious that compared to DIW, how aerosol jet would perform when printing aerogel, e.g. distribution of CNCs within the PI. See below a related paper. Could the authors include it in the discussion?
- Goh, Guo Liang, Shweta Agarwala, and Wai Yee Yeong. "Aerosol-jet-printed preferentially aligned carbon nanotube twin-lines for printed electronics." ACS applied materials & interfaces46 (2019): 43719-43730.
Reviewer 2 Report
- Thank you for submitting your paper. The work done here draws attention to a significant subject. I have found the paper very interesting. The article is well written and well structured as a scientific text. However, several issues need to be addressed properly before the paper is being considered for publication. My comments including major and minor concerns are given below:
- Please consider reviewing the abstract and highlight the novelty, major findings and conclusions. I suggest reorganizing the abstract, highlighting the novelties introduced. It should contain answers to the following questions:
- What problem was studied and why is it important?
- What methods were used?
- What conclusions can be drawn from the results?
- What is the novelty of the work and where does it go beyond previous efforts in the literature?
- Just before the last paragraph in the introduction, the authors should answer the following question: What is the research gap did you find from the previous researchers in your field? Mention it properly. It will improve the strength of the article.
- Materials and methods section is comprehensive and clear, however, images and graphs of equipment used, samples fabricated, and tests implemented with details on those images should be provided, this is an experimental study, and it is important to give sufficient information to the readers about the work done here.
- Line 184 why the “gel behaviour” occurred at low shear stress, it is not enough to state an observation or a finding without a) explaining the phenomena and the reason behind it (supported with references) and b) comparing your findings with past studies similar or closely related to your work. This must be done in every place where the authors state a new observation/claim or finding.
- Perhaps a more descriptive title that better highlights the work done should be used. The title of the manuscript is somewhat generic and does now tell the readers what exactly was done in it.
- Please avoid using we, us or our, please check this issue everywhere in the manuscript.
- Lines 206-208 please support these claims with a reference(s).
- 3.2. Chemical analysis of the PI/CNCs nanocomposite aerogels section is well discussed and detailed, this is what should be done to the other sections as well.
- I strongly suggest remove the discussion section as it is too short and just combine it with the results section and rename it as results and discussion section.
- The results are somewhat described and is limited to comparing the experimental observation. The authors are encouraged to include more detailed discussion and critically discuss the observations from this investigation with existing literature.
- Conclusion is weak, please expand upon and use bullet points if possible. (1-2 bullet points for each of the subsections in the results and discussion sections.
